# Manipulation of Subwavelength Periodic Structures Formation on 4H-SiC Surface with Three Temporally Delayed Femtosecond Laser Irradiations

**DOI:** 10.3390/nano12050796

**Published:** 2022-02-26

**Authors:** Wanlin He, Bo Zhao, Jianjun Yang, Junqing Wen, Hua Wu, Shaoli Guo, Lihua Bai

**Affiliations:** 1School of Science, Xi’an Shiyou University, Xi’an 710065, China; wlhe@xsyu.edu.cn (W.H.); jqwen1221@xsyu.edu.cn (J.W.); whua@xsyu.edu.cn (H.W.); guoshaoli@mail.nwpu.edu.cn (S.G.); lhbai@xsyu.edu.cn (L.B.); 2Changchun Institute of Optics, Fine Mechanics and Physics, Chinese Academy of Sciences, Changchun 130033, China; zy828522@163.com; 3Department of Electronic Information and Physics, Changzhi University, Changzhi 046011, China

**Keywords:** femtosecond laser, laser micromachining, surface plasmon polarization, micro/nanostructure, transient temperature grating, ultrafast surface dynamics

## Abstract

Controlling laser-induced periodic surface structures on semiconductor materials is of significant importance for micro/nanophotonics. We here demonstrate a new approach to form the unusual structures on 4H-SiC crystal surface under irradiation of three collinear temporally delayed femtosecond laser beams (800 nm wavelength, 50 fs duration, 1 kHz repetition), with orthogonal linear polarizations. Different types of surface structures, two-dimensional arrays of square islands (670 nm periodicity) and one-dimensional ripple structures (678 nm periodicity) are found to uniformly distribute over the laser-exposed areas, both of which are remarkably featured by the low spatial frequency. By altering the time delay among three laser beams, we can flexibly control the transition between the two surface structures. The experimental results are well explained by a physical model of the thermally correlated actions among three laser-material interaction processes. This investigation provides a simple, flexible, and controllable processing approach for the large-scale assembly of complex functional nanostructures on bulk semiconductor materials.

## 1. Introduction

Femtosecond laser-induced periodic surface structures (fs-LIPSSs), as universal phenomena that occur on all types of materials including metals, semiconductors, and dielectrics [1,2,3], have been a topic of intensive study for decades. The fs-LIPSS technology, with a mask-free and single-step process, represents a superior micro- and nanoscale engineering technology for material surfaces over several top-down micro/nanofabrication techniques, including photon/electron-beam lithography [4,5], nanoimprinting [6], and laser direct writing [7], which often suffer from having high-cost, time-consuming, or chemical procedures [8]. The fs-LIPSS techniques with subwavelength or nanometer scales have the capability of manipulating the optical, mechanical, wetting, chemical, biological, and other properties of material surfaces, providing wide applications in many fields such as enhanced optical absorption [9], plasmonics [10], magnetic recording [11], sensing [12], and metal coloring [13]. Generally, the processing of LIPSSs is strongly dependent on laser conditions (wavelength, pulse energy, number of pulses, and the polarization state), material properties, and ambient environment [13,14,15].

One-dimensional (1D) LIPSSs, often termed ripples, formed under irradiation of a single femtosecond laser beam, have less degree of freedom for manipulating the optical, mechanical, wetting, chemical, biological and other properties, thus limiting the diversity of surface functionalizations. In general, 1D ripple structures often present two features: high spatial frequency (HSF) and low spatial frequency (LSF). Additionally, the switching from LSF to HSF cases can be tuned by a single laser pulse irradiation on semiconductors and dielectrics [16,17,18], but its controlling flexibility is very poor, which is mainly manifested as a self-organizing behavior. To expand the scope of surface functionalizations for multidisciplinary applications, some attempts have been made to fabricate two-dimensional (2D) LIPSSs, which mainly use either the double exposure with two laser beam interference or the single exposure with multi-beam interference [19,20]. Although these methods can produce large-area 2D LIPSSs, they are usually subjected to complexity for the design and adjustment of experimental optical paths.

Recently, double femtosecond laser irradiations with certain temporal delays were proposed to flexibly control the morphology of LIPSSs via the transient surface properties of the optically excited materials. For example, through using double-fs-laser-beam irradiation, two kinds of 1D LIPSSs—high-spatial-frequency (HSF) and low-spatial-frequency (LSF) ripple structures—were demonstrated on silicon surfaces at different time delay regimes [21,22,23,24]. That is, the spatial periods of LIPSSs on silicon surfaces can be tuned via varying the time delay between double laser beams. The recent investigations of such a technique on different metal surfaces (tungsten, molybdenum, and nickel) and semiconductors (silicon carbide, diamond), or even transparent materials, demonstrated that several new types of 2D-LIPSSs, consisting of nanobump, triangular, and dot arrays, have been successfully created [25,26,27,28]. Furthermore, triangular 2D-LIPSSs, square 2D-LIPSSs, and round structures created on the surface of stainless steel via double-pulse irradiation were previously reported [29]. In general, by altering the time delay between double-fs-laser beams, the surface morphology of 2D-LIPSSs, such as geometrical profiles, spatial periods, and structure feature sizes, can be flexibly tailored. In addition, concentric circular millimeter-scale macrostructures with a tail can be produced through femtosecond laser micro/nano dual-scale composite texture technology, as well as concentric circular structures with low/high/ultra-periodic nanostructures [30,31].

It should be pointed out that the formation of 2D-LIPSSs under irradiation of double-fs-laser-beam on semiconductor surfaces was rarely reported, and the morphology distribution of 1D ripple surface structures mostly presented low regularity. In our recent report of a three-fs-laser-beam irradiation experiment, the regularity of 1D ripple structures formed on a 4H-SiC surface was significantly improved, and the structure period was found to transfer from HSF to LSF regime [32], which performs more degrees of freedom and has superiority in controlling the LIPSS formation.

In this paper, we investigate the control of 2D LIPSS formation on a 4H-SiC crystal surface by using the temporally delayed irradiation of triple femtosecond laser beams, associated with the orthogonal linear polarizations in successive sequences. When their mutual time delays are varied in a range of 0~60 ps, we can observe the formation of 1D ripple structures and 2D periodic arrays of square islands, both of which possess the evident LSF features and the uniform distribution, i.e., a larger spatial period similar to that occurring in a metal-like state. Subsequently, we provide discussion and analyses based on mutual correlations among three ultrafast surface dynamics, which gives new insights into the nonequilibrium change in the material during the laser-induced structures.

## 2. Experimental

A schematic diagram of the experimental setup is shown in Figure 1, where a Ti: femtosecond laser amplifier system (SpectraPhysics HP-Spitfire 50) was adopted as the light source, to deliver the horizontally polarized pulse trains (*f* = 1 kHz, *λ* = 800 nm, *τ* = 50 fs), with the maximum energy of 2 mJ. At first, each laser pulse out of the amplifier was divided into three subpulse beams (P_1_, P_2_, and P_3_) via two splitters (BS_1_ and BS_2_). During the optical paths of P_2_ and P_3_ beams, we employed two delay lines for producing the temporal delays among three subpulses. Moreover, two half-wave plates were also inserted into the optical paths of P_1_ and P_3_ beams to change their linear polarization from the horizontal to the vertical directions. Afterward, three laser beams with mutual time delays were spatially realigned into a collinear propagation and then focused by an objective lens (4×, N. A = 0.1) onto the sample surface at normal incidence. A single-face polished crystal of 4H-SiC (with dimensions of 10 mm × 10 mm × 1 mm), which has potential applications in high-temperature electronic devices, was selected as the sample material in the experiment, and it was mounted on a computer-controlled, three-dimensional (3D) translation stage. In order to avoid the strong ablation damage, the sample was placed at the position near 300 μm away before the focus, leading to a beam spot diameter of *ϕ* = 60 μm on the sample surface. Under the fixed irradiation of femtosecond laser pulses, a line-scribing method was carried out by translating the sample at the speed of *v* = 0.1 mm/s, resulting in 600 laser pulses partially overlapped within the spot area. Before and after the experiments, the sample surface was ultrasonically cleaned in acetone solution for 30 min. Morphologies of the laser-exposed surface were characterized by both scanning electron microscopy (SEM) and atomic force microscopy (AFM). In the experiment, laser beams of P_1_ and P_3_ were linearly polarized in the vertical direction, while P_2_ was polarized along the horizontal direction. For simplicity, Δt_1_ represents the time delay between P_1_ and P_2_ beams, and Δt_2_, the time delay between P_2_ and P_3_ beams.

## 3. Results and Discussion

For the sake of comparison, we firstly investigated the morphologies of the sample surface under irradiation of the single-beam femtosecond laser, by separately using the individual laser beam P_1_, P_2_, or P_3_ at the identical energy fluence of 0.24 J/cm^2^, and the corresponding results are shown in Figure 2. Similar to previous observations [33,34], the HSF-type ripple structures are found to generate on laser-irradiated surfaces whose profile consists of multiple short-ranged fragments with the semi-periodic interspacing of approximately Λ = 150 nm, less than half the wavelength of the incident light. Moreover, for the incident laser with different linear polarizations, the spatial orientation of ripple structures becomes varied, but it is always perpendicular to the laser polarization direction. Notably, while keeping the case of the single laser beam irradiation, the abovementioned semi-periodic surface structures will no longer be formed on the 4H-SiC crystal surface, if laser energy fluence is attenuated to F = 0.07 J/cm^2^, below the ablation threshold (0.14 J/cm^2^).

More interestingly, when irradiation of the single laser beam at such a low energy fluence changed into three light beams (0.07 J/cm^2^/pulse), especially with the time delays of Δt_1_ = 20 ps and Δt_2_ = 30 ps, it was surprisingly found that a new type of the periodic structures can be formed on the sample surface that displays the geometric profiles of 2D square island arrays, as shown in Figure 3a. In startling contrast to the aforementioned 1D HSF ripple structures, such kind of structure formation undoubtedly implies physical correlations among dynamic processes of three laser-material interactions. The peculiarities of such surface structures can be described as follows: (1) the spatial alignments of the square islands seem to be periodic in two directions that are, respectively, parallel or perpendicular to the incident laser polarization; (2) the measured periods of the structures in two (both vertical and horizontal) directions are almost equal to Λ = 670 nm, which belong to the typical LSF phenomenon; (3) the characteristic dimension of the structure cell is as small as about D = 486 nm.

Figure 3b,c present am AFM topography of the 2D nanoisland structures and a measured curve of the cross-section, respectively, which indicate the average modulation depth of approximately H = 175 nm. Moreover, a fast Fourier transformation of the corresponding surface structures was also calculated, as shown in Figure 3d, wherein the periodically distributed tiny bright spots in the frequency domain also imply the highly regular distribution of the laser-induced surface structures. The measured interval between adjacent spots is about *f* = 1.49 μm^−1^, whose corresponding reciprocal is equal to the measured structure period from SEM images. It should be stressed here again that the formation of such 2D LSF island structures by three laser beams is clearly much different from the previous observations on surfaces of bulk semiconductor and dielectric materials with irradiation of single-beam femtosecond laser, the latter of which usually exhibit the irregular distribution of the deep-subwavelength-scaled structures [35]. Instead, the attained structure period close to the incident laser wavelength is more similar to the observations on the metal surfaces [36], even though the two materials have many physical properties.

Subsequently, we investigated how the time delay of triple laser irradiations (0.07 J/cm^2^/pulse) affects the structure formation on 4H-SiC material. For example, when the first delay time increased up to Δt_1_ = 40 ps while keeping the second delay time at Δt_2_ = 30 ps, it is unexpectedly found that the formation of the surface structures turn to present 1D periodic grating profiles rather than 2D periodic arrays of square islands, as shown in Figure 4. In this case, the formation of 1D grating-like structures appears to be uniform, with the measured spatial period of about Λ = 678 nm, still belonging to the LSF phenomenon. Remarkably, the spatial orientation of 1D ripple structures becomes perpendicular to the polarization direction of the third subpulse laser irradiation, P_3_. The measured duty ratio of the grating ridge to the structure period approximates Υ = 0.71 (Υ = D/Λ, where D is the width of the grating ridge), which indicates the ablation groove width is about W = 203 nm, and the HSF ripple structures usually have the value of Υ = 0.9. This result is much different from situations obtained upon laser irradiation of either the single beam or double beams. For the case of the double-fs-beam irradiation, 1D grating-like ripple structures with HSF features were induced on the 4H-SiC crystal surface by using only two time-delayed femtosecond laser pulses with different linear polarizations, and it has been observed that the available ripple orientation also tended to change with different time delays, but the ripple period almost stayed invariant of approximate Λ = 150 nm [37].

Figure 5 shows the evolution of surface morphology induced by triple laser (0.07 J/cm^2^/pulse) irradiations on the 4H-SiC surface with the second time delay variation within a range of Δt_2_ = 18~32 ps, when the first time delay was fixed at Δt_1_ = 20 ps and Δt_1_ = 40 ps, respectively. Clearly, in the case of Δt_1_ = 20 ps, LSF-typed 2D periodic arrays of the square structures are always generated with varying the second time delay Δt_2_; conversely, in the case of Δt_1_ = 40 ps, LSF-typed 1D periodic ripple structures can be produced regardless of changing the second time delay Δt_2_, with orientation perpendicular to the polarization direction of the third subpulse laser irradiation P_3_. These phenomena imply that the first time delay Δt_1_ should play a predominant role in controlling the structural morphology on 4H-SiC surfaces. Figure 6 displays the measured dependence of either the spatial period or the duty ratio of 1D LSF ripple structures on the second time delay Δt_2_ under the condition of Δt_1_ = 40 ps. It can be seen that during the large variation range of Δt_2_ = 4~60 ps, both structural parameters stayed almost unchanged.

To elucidate the physical origins of the aforementioned two different types of LSF structures on the crystal 4H-SiC surfaces under triple temporally delayed femtosecond laser irradiations, we need to consider the ultrafast change in surface properties of the material. According to the previous studies [38,39], upon femtosecond laser irradiation, a thin layer of plasma is transiently produced on the semiconductor surface due to the laser-excited free electrons via multiphoton absorption, so that the dielectric permittivity of the material is tuned by the formula of $\varepsilon_{m}=1-\frac{\omega_{p}{}^{2}}{\omega^{2}+\gamma^{2}}$, where *ω_p_* is the electron density (*N_e_*) dependent plasma frequency, *ω* for the incident laser frequency, and *γ* for the electron collision frequency. If the electron density is above the critical value of *N_e_*th (corresponding to $\varepsilon_{m}=-1$), the laser-exposed material surface is switched into a plasmonically active condition, which allows a direct excitation of SPP similar to a metallic state. Under the irradiation of single-beam laser pulses, the formation of 1D HSF-typed ripple structures is physically originated from interference between the incident laser and its excited surface plasmon polaritons [37], which tends to modulate the continuous distribution of the Gaussian laser energy into the intensity fringes, resulting in the spatially periodic optical absorption and heating of the electrons. After electron–phonon coupling and thermal relaxation within the time scale of tens of picosecond [40], a grating-like distribution of the material temperature is established, i.e., the so-called transient temperature grating, which finally evolves into the permanent periodic ablation patterns via selective material removal once it is above the damage threshold temperature of Tth. The resultant ripple period can be estimated by Λ = *λ_0_*|1/*ε_d_* + 1/*ε_m_*|^1/2^ [15], where *ε_d_* is the permittivity of the ambient medium of the laser-excited sample surface. It is clear that the spatial period of 1D ripple structures formed on the 4H-SiC surface is dependent on the density of laser-excited free electrons. Normally, under the single-beam irradiation, the medium surrounding the material surface is air substance, i.e., *ε_d_* = 1, and the value of (1/*ε_d_* + 1/*ε_m_*)^1/2^ is far less than 1, which consequently creates the HSF-typed periodic surface structure.

Based on the above discussion, we now propose a physical scenario to explain the structure formation on the 4H-SiC surface when irradiated by three temporally delayed femtosecond lasers, as sketched in Figure 7, in which each curve indicates the temporal evolution of the temperature of the transient grating induced by the individual subpulse laser irradiation. Noticeably, the maximum temperature of the transient grating induced by each subpulse laser irradiation should be below the threshold value Tth, because of having sufficiently low energy fluence. For the shorter time delay of Δt_1_ = 20 ps, because the surface temperature heated by the first laser irradiation slightly decays from the peak into a certain level, it is possible for the second laser irradiation to promote the transient grating temperature larger than the threshold value of Tth, which eventually makes the periodic ripple structures permanently imprint on the material surface, whose orientation is perpendicular to the polarization direction of the second laser irradiation. Similarly, when the third laser irradiation arrives during the proper time delay range of 18 ps ≤ Δt_2_ ≤32 ps with respect to the second laser irradiation, its transient grating temperature is also feasible to exceed the threshold value of Tth because the material is characterized by the residual high temperature after preceding two laser irradiations. As a consequence, another group of the periodic ripple structures can be imprinted on the material surface, with orientation perpendicular to the polarization direction of the third laser irradiation. Moreover, because the second and the third laser irradiations are linearly polarized in orthogonal directions, the spatial arrangements for their transient gratings or imprinting ripple structures cross each other, resulting in 2D arrays of square islands. Remarkably, compared with the case of single-beam laser irradiation, the two interlocking ripple structures induced by triple femtosecond lasers are seen to have the typical LSF features in the spatial period, which can be understood as follows: because the ripple period formation on the sample surface is usually varied by free electrons on laser-exposed area, the latter of which can reach higher density after the second and third laser irradiations, the resultant smaller negative permittivity of *ε_m_* certainly leads to LSF-type surface structures.

On the other hand, when the first time delay increased to Δt_1_ = 40 ps, the target surface temperature heated by the first laser irradiation is ready to decay into a significantly low temperature via thermal diffusion before the second laser’s arrival. Subsequently, the transient grating temperature induced by the second laser irradiation fall below the threshold of Tth due to the weaker preheating contribution of the first laser, which degrades the dominant role in imprinting the ripple-like surface morphology. During the proper time delay range of 58 ps ≤ Δt ≤72 ps (i.e., Δt = Δt_1_ + Δt_2_), the transient grating temperature heated by the third laser irradiation is feasible to exceed the threshold value of Tth on the nonequilibrium surface due to the preheating contributions of the first two laser irradiations, which will uniquely predominate the surface morphology change, leading to the formation of 1D grating-like ripple structures with orientation perpendicular to the polarization direction of the laser irradiation P_3_. It is clear that the period of 1D ripple structures also have the LSF-type period because the dielectric permittivity of the 4H-SiC surface tends to become smaller after the second and third laser irradiations, and the value of the item (1/*ε_d_* + 1/*ε_m_*)^1/2^ will increase to a value close to 1, which leads to the LSF period of LIPSSs. Therefore, the formation of uniform LSF-type ripple structures indicates that the physical correlations among three dynamic processes of femtosecond laser–material interaction benefit the excitation of the free electrons to modulate the structure’s formation.

## 4. Conclusions

In conclusion, we demonstrated, for the first time, that the uniform LSF-type structures can be produced on 4H-SiC semiconductor surfaces upon irradiation of three temporally delayed femtosecond laser beams, much different from the common observations of HSF-type structures induced by the single-beam laser irradiation. The morphology of the surface structures can be modified into two different profiles: 1D periodic ripple structures and 2D periodic arrays of square islands, in which the time delays among three laser irradiations were varied. A physical model relevant to the SPP-based transient temperature grating was proposed to well explain the formation of such structures, in which the dielectric permittivity of the material surface was modulated into a metal-like state, consequently resulting in the larger spatial period in these structures. Our investigations reveal that using temporally delayed incident femtosecond laser beams can effectively control surface morphology and can also make a significant improvement in structural regularity via the physical correlations of ultrafast surface dynamics, providing potential applications in femtosecond laser processing of metamaterial and plasmonic devices.

## Figures and Tables

**Figure 1 nanomaterials-12-00796-f001:**
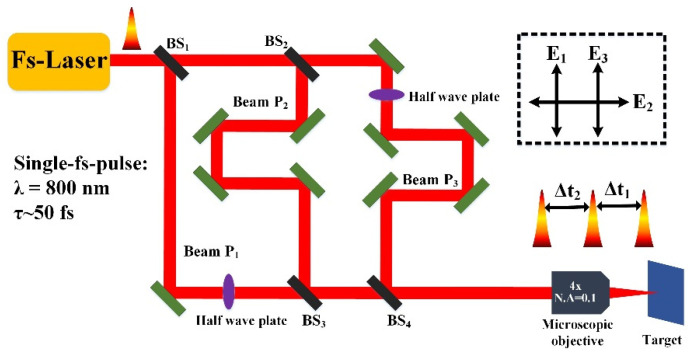
Schematic diagram of the experimental setup. BS_1_, BS_2_, BS_3_, and BS_4_ represent beam splitters; fs, femtosecond; an inset diagram illustrates the polarization directions of three laser irradiations of P_1_, P_2_, and P_3_. E_1_, E_2_, E_3_ represent the electric field of three laser irradiations of P_1_, P_2_, and P_3_.

**Figure 2 nanomaterials-12-00796-f002:**
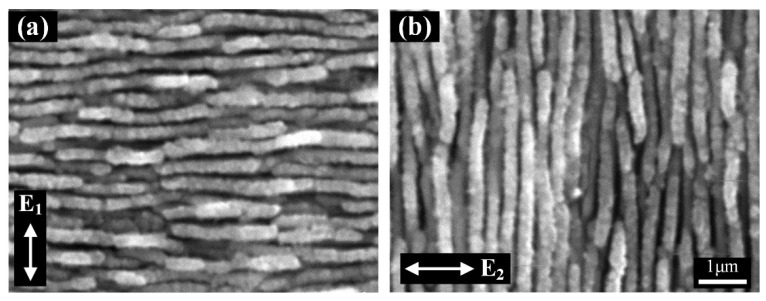
(**a**,**b**) SEM images of the HSF ripple structures formed on the surface of 4H-SiC crystal with the individual femtosecond laser irradiation of P_1_ (or P_3_) and P_2_, respectively, at the energy fluence of F = 0.24 J/cm^2^. Here, the white double solid arrows denote the polarization directions of the laser irradiations.

**Figure 3 nanomaterials-12-00796-f003:**
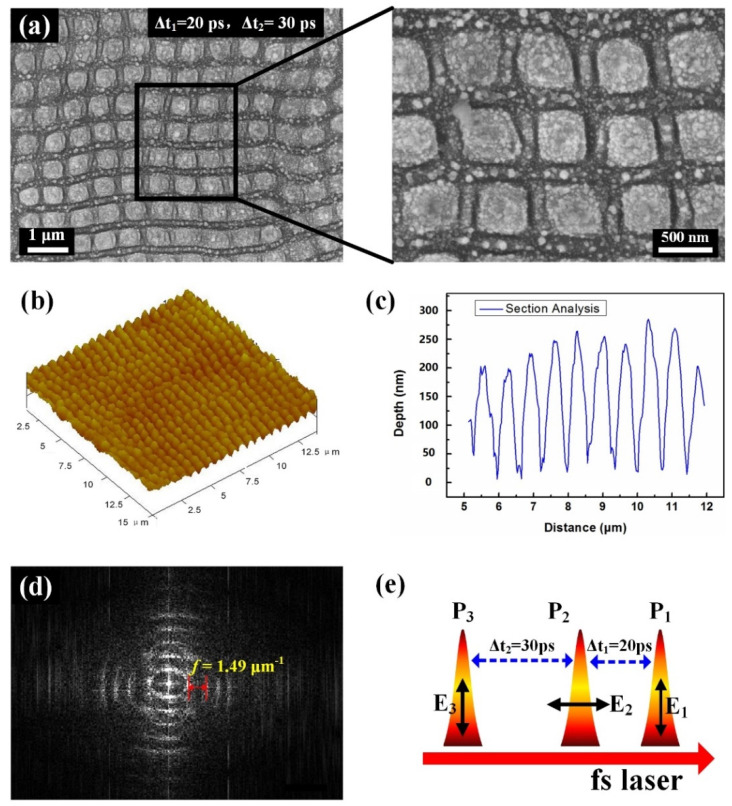
(**a**) SEM images of 2D arrays of LSF-type square island structures formed on the surface of 4H-SiC crystal using three collinear femtosecond laser irradiations that are mutually time-delayed by Δt_1_ = 20 ps and Δt_2_ = 30 ps. Here, the energy fluence of each laser beam is F = 0.07 J/cm^2^, and the polarization directions of three laser irradiations are denoted by the double arrows; (**b**,**c**) AFM image of the 2D arrays of LSF-type square island structures and the measured cross-section profile; (**d**) image of the fast Fourier transformation (FFT) of (**a**); (**e**) diagram of pulse time sequence among the three laser beams. E_1_, E_2_, E_3_ illustrate the polarization directions of three laser irradiations of P_1_, P_2_, and P_3_.

**Figure 4 nanomaterials-12-00796-f004:**
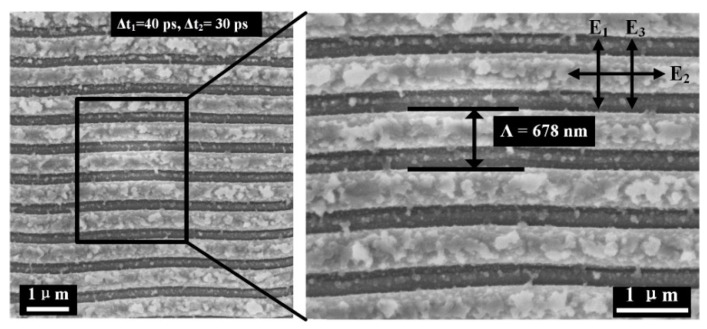
SEM image of 1D LSF-type ripple structures on 4H-SiC crystal surface using three femtosecond laser irradiations with the time delay of Δt_1_ = 40 ps and Δt_2_ = 30 ps, where the energy fluence of each laser beam is F = 0.07 J/cm^2^, and the polarization directions of the three laser irradiations are mutually orthogonal.

**Figure 5 nanomaterials-12-00796-f005:**
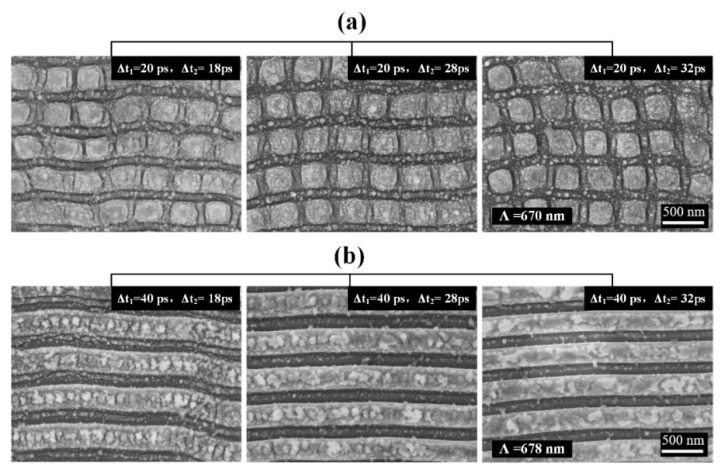
SEM images of the LSF-type surface structures formed on 4H-SiC crystal using three temporally delayed collinear femtosecond laser irradiations with orthogonal polarizations, where the energy fluence of each laser irradiation is F = 0.07 J/cm^2^: (**a**) two-dimensional (2D) arrays of square island structures at Δt_1_ = 20 ps, 18 ps < Δt_2_ < 32 ps; (**b**) one-dimensional (1D) LSF ripple structures at Δt_1_ = 40 ps, 18 ps < Δt_2_ < 32 ps.

**Figure 6 nanomaterials-12-00796-f006:**
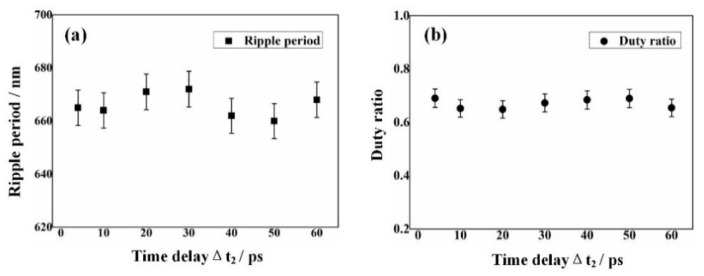
(**a**,**b**) Measured spatial period and duty ratio of 1D LSF-type ripple structure as a function of the second time delay Δt_2_, respectively, when the time delay Δt_1_ is fixed at 40 ps.

**Figure 7 nanomaterials-12-00796-f007:**
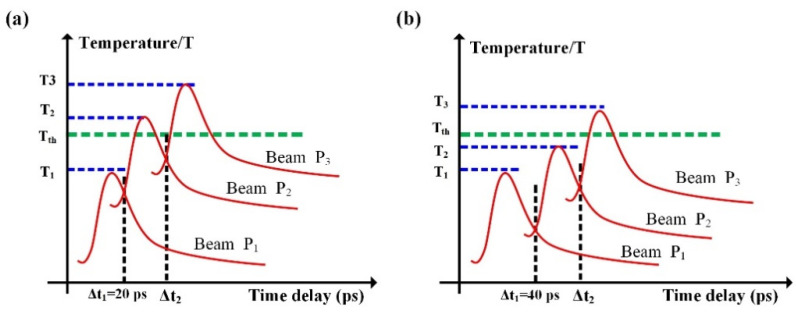
(**a**,**b**) Schematic diagram of the temporal evolution of the temperature of three transient gratings *k_ig_* (*i* = 1, 2, 3) that are excited by three temporally delayed femtosecond laser irradiations.

## Data Availability

The data presented in this study are available in this article.

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
