# Peer review of "Manipulation of Subwavelength Periodic Structures Formation on 4H-SiC Surface with Three Temporally Delayed Femtosecond Laser Irradiations"

_nanomaterials, 2022, doi:10.3390/nano12050796_

Round 1

Reviewer 1 Report

The experiments follow a solid scientific approach and the results are presented in an accurate and logic manner. Therefore the manuscript can be accepted without further changes.

Author Response

Responses to Referees Comments

We would like to thank the editor for giving us a chance to resubmit the paper, and also appreciate the referees for their insightful suggestions. The following is a point-by-point response to the three reviewers’ comments.

Reviewer #1

The experiments follow a solid scientific approach and the results are presented in an accurate and logic manner. Therefore the manuscript can be accepted without further changes.

Response: Thanks very much for your comments.

Reviewer 2 Report

Report on the manuscript titled “Manipulation of subwavelength periodic structures formation on a 4H-SiC surface with triple delayed femtosecond laser irradiation” by W. He et.al

In this paper, the authors demonstrate the control of periodic structures (2D LIPPS in particular) formation on 4H-SiC crystal surface using a femtosecond laser beam delayed temporarily from 0 to 60 ps. The creation of the 2D square structures via the usage of delayed femtosecond laser pulses is novel and the manuscript is written in a scientific manner. Therefore, I recommend the acceptance of the paper after addressing the following points

  • What is the role of scanning speed in the formed structures? The results are presented here for only one scanning speed. How do the structures get affected by different scanning speeds?
  • Whether the interchange of time delays i.e and  result in similar structures?
  • Give a little more detailed discussion on how critical the polarization of each laser beam is on the forming periodic structures
  • Whether the researchers are attempted similar studies on any other substrate so that postulated surface temperature variation can be compared?

Author Response

Responses to Referees Comments

We would like to thank the editor for giving us a chance to resubmit the paper, and also appreciate the referees for their insightful suggestions. The following is a point-by-point response to the three reviewers’ comments.

Reviewer #2

Report on the manuscript titled “Manipulation of subwavelength periodic structures formation on a 4H-SiC surface with triple delayed femtosecond laser irradiation” by W. He et.al. In this paper, the authors demonstrate the control of periodic structures (2D LIPPS in particular) formation on 4H-SiC crystal surface using a femtosecond laser beam delayed temporarily from 0 to 60 ps. The creation of the 2D square structures via the usage of delayed femtosecond laser pulses is novel and the manuscript is written in a scientific manner. Therefore, I recommend the acceptance of the paper after addressing the following points

  • What is the role of scanning speed in the formed structures? The results are presented here for only one scanning speed. How do the structures get affected by different scanning speeds?

Response: In our paper, the experiment was carried out by translating the sample at certain speeds, and this method has been already demonstrated in many research groups including our previous study [S1-S3]. The line-scanning speed is a key parameter in the formation of uniform periodic ripple structures on material surface. Specifically, the line-scribing process allows a partial overlapping of a number of laser pulses at given areas, and the line-scanning speed will determine how many pulses overlapping in the area of laser irradiation on the sample surface, which eventually affects the information of surface structures (especially the regular arrangement).

In this manuscript, a line-scribing method was carried out at the speed of v = 0.1 mm/s, resulting in 600 laser pulses partially overlapped within the spot area. In fact, the experiments at different scanning speeds (v = 0.01, 0.05, 0.1, 0.2, 0.3 mm/s) have been performed in our research to obtain the periodic surface ripple structures. However, the ripple surface structures were found to have the optimal arrangement (uniform and clear) at the speed of v =0.1 mm/s, and the spatial regularity of the ripple structures appeared to be worse when the scanning speed became too large or too small. Therefore, we here only selected the results under the condition of one scanning speed.

[S1] Öktem, B.; Pavlov, I.; Ilday, S.; KalaycıoÄŸlu, H.; Rybak, A.; Yavas, S.; ErdoÄŸan, M.; Ömer Ilday, F. Nonlinear laser lithography for indefinitely large-area nanostructuring with femtosecond pulses. Nat. Photon. 2013, 7, 897-901.

[S2] Ruiz de la Cruz, A.; Lahoz, R.; Siegel, J.; Francisco de la Fuente, G.; Solis, J. High speed inscription of uniform, large-area laser-induced periodic surface structures in Cr films using a high repetition rate fs laser. Opt. Lett. 2014, 39, 2491-2494.

[S3] He, W.; Yang, J.; Guo, C. Controlling periodic ripple microstructure formation on 4H-SiC crystal with three time-delayed femtosecond laser beams of different linear polarizations. Opt. Express 2017, 25, 5156.

  • Whether the interchange of time delays i.e and result in similar structures?

Response: In our experiment, we have investigated the formation of the surface structures by changing the time delays (Δt1 = 20 ps, Δt2 = 0-60 ps; Δt1 = 40 ps, Δt2 = 0-60 ps) between the three pulses. However, the results revealed that the evolution of the surface structures with the time delays can be categorized into two types: one is identical to the observations of the traditional ripple structures (at Δt1 = 20 ps), and the other is similar to the observations of the square-like structures (at Δt1 = 40 ps), which are represented by Figures. 3a and 4, respectively. Therefore, in our manuscript, the first time delay of Δt1 plays an important role in the formation of these two types of surface structures. In fact, the interchange of time delays were included in our experiment, so the same results still occur if the time delays exchanged.

  • Give a little more detailed discussion on how critical the polarization of each laser beam is on the forming periodic structures

Response: The polarization is an extremely important laser parameter, to remarkably affect femtosecond laser induced periodic surface structures (LIPSSs). That is to say, the spatial orientation and profiles of the nanostructures depends on the incident laser polarization. In general, numerous experimental data have shown that the periodic ripple structures can be induced by using linearly polarized ultrashort laser pulses, and ripples have been found to grow on the surface, with orientation perpendicular to the polarization of the incident beam. For the case of three time-delayed femtosecond laser beam irradiations, when the sample surface is irradiated by the first laser pulse with the horizontal polarization, a thin layer of plasma is transiently produced on the semiconductor surface due to the laser-excited free electrons via multiphoton absorption, which allows a direct excitation of SPP-like metallic state, resulting in the spatially periodic optical absorption and heating of the electrons. After electron-phonon coupling and thermal relaxation within the time scale of tens of picosecond, a grating-like distribution of the material temperature is established, i.e., the so-called transient temperature grating with a grating vector k1g parallel to the first laser polarization. The permanent periodic patterns are ablated via the selective material removal once the temperature is above the damage threshold of Tth.

When the second laser pulse of P2 with the vertical polarization is incident during the relaxation processes of the transient temperature grating, a transient temperature grating with the grating vectors k2g parallel to the second laser polarization is also established and it couples to the first transient temperature grating of k1g. For the shorter time delay of Δt1 = 20 ps, because the surface temperature heated by the first laser irradiation slightly decays from the peak into a certain level, it is possibly for the second laser irradiation to promote the transient grating temperature larger than the threshold value of Tth, which eventually makes the periodic ripple structures permanently imprint on the material surface, whose orientation is perpendicular to the direction of the second laser polarization. Similarly, we can consider that the above physical process also provides a new transient temperature grating of k3g for the time-delayed incident third femtosecond laser pulse of P3 that is linearly polarized in the horizontal direction.

In this case, its transient grating temperature is also feasible to exceed the threshold value of Tth, because the material is characterized by the residual high temperature after the preceding two laser irradiations. As a consequence, another group of the periodic ripple structures can be imprinted on the material surface, with orientation perpendicular to the direction of the third laser polarization. Moreover, because the second and the third laser pulse irradiations are linearly polarized in orthogonal directions, the spatial arrangements for their transient gratings or imprinting ripple structures become crossed each other to result in 2D arrays of square islands.

Based on the above analysis, we can find that the polarizations of femtosecond laser pulses play an important role in the formation of periodic surface structures on material surfaces.

  • Whether the researchers are attempted similar studies on any other substrate so that postulated surface temperature variation can be compared?

Response: In fact, the effect of triple temporally delayed femtosecond laser pulses on the formation of 2D LIPSSs on a nickel (Ni) surface was already performed in our previous works [S4]. The 2D structures are fabricated over the entire irradiated region with relatively high uniformity. The experiment result showed that using a triple-pulsed beam allows for additional degrees of freedom that can enable the control over different structural parameters, namely, the period and structures dimensions. We believe that the variation of surface temperature is an important factor in the formation of 2D structures on nickel surface.

[S4] Jalil, S.A.; Yang, J.; Elkabbash, M.; Lei, Y.; He, W.; Guo, C. Formation of uniform two-dimensional subwavelength structures by delayed triple femtosecond laser pulse irradiation. Opt. let. 2019, 9, 2278-2281.

Reviewer 3 Report

This paper by He et al. reports the fabrication of bi-dimensional (2D) nanostructures using three collinear and temporally delayed femtosecond (fs) laser pulses. Additionally, the authors demonstrate that it is possible to control the structures' geometry by altering the mutual time delays among them.

The work is interesting and will be appreciated by the scientific community, nevertheless, it contains some paragraphs that are not written in a clear way and needs to be reworded before publication (e.g. page 2 - lines 63-65: “Especially, during these double-fs-laser-beam irradiation experiments on semiconductor surfaces, the formation of 2D-LIPSSs has a rarely report, and the morphology distribution of 1D-ripple surface structures mostly presents the low regularity”.

Overall, the work is remarkable particularly in terms of the presented physical model and the possible technological impact of the nanostructures. I suggest publication only after addressing the following points:

1) Important information is missing in the abstract. The authors should provide

relevant measurements about the structures morphology (e.g. periodicity) and also laser and processing parameters should be specified, (e.g. wavelength, pulse duration, repetition rate). Since in the introduction it is emphasized how this technique could replace electron-beam lithography, it is important to provide the reader with information about the periodicity of the structures.

2) From the introduction it seems to be implied that the transition from HSL- to LSF- LIPSSs is possible only when using a double pulse irradiation, however the transition from fine to coarse ripples has already been reported both for 6H-SiC and 4H-SiC by: 1) S.H. Kim, K.H. Byun, I.B. Sohn, S.H. Jeong, Progressive formation of fine and coarse ripples on SiC surface by repeated irradiation of femtosecond laser pulses, Applied Physics B 113 (2013) 395-402; 2) T. Tomita, K. Kinoshita, S. Matsuo, S. Hashimoto, Distinct Fine and Coarse Ripples on 4H-SiC Single Crystal Induced by Femtosecond Laser Irradiation, Japanese Journal of Applied Physics 45 (2006) L444-L446.

For the importance of LIPSS of such systems and describe the significant differences in scales of the created nanostructures and what laser parameters might affect the self-organization process it is strongly suggested to consider the following very recent paper published and References reported therein: Mastellone M. et al., LIPSS Applied to Wide Bandgap Semiconductors and Dielectrics: Assessment and Future Perspectives, Materials 15 (2022) 1378.

It should be stressed that the switch from LSF to HSF can be tuned by even a single pulse irradiation.

3) In the introduction are reported examples of regular 2D structures obtained on metal surfaces (tungsten, molybdenum and nickel). It is important to also show that it is indeed possible to obtain 2D structures on semiconductors or even transparent (non-absorbing) materials controlling the spatial period and the geometrical patterns of LIPSSs: 1) Wu, X., Jia, T., Zhao, F. et al. Formation mechanisms of uniform arrays of periodic nanoparticles and nanoripples on 6H-SiC crystal surface induced by femtosecond laser ablation, Applied Physics A 86, 491-495 (2007), 2) Mastellone, M. et al., Deep-Subwavelength 2D Periodic Surface Nanostructures on Diamond by Double-Pulse Femtosecond Laser Irradiation, Nano Letters 21 (2021) 4477-4483.

By reporting these works it could be explained why a three-pulse sequence is preferable and how the formation mechanism differs.

4) In the manuscript it is not that clear that the energy fluence used for the 1-pulse experiment (0.24 J/cm2) is different than the one used the 3-pulse irradiation (0.07 J/cm2). This is just reported in the caption of Fig.3 and it should be clearly explicated in the text.

5) Figures 3a and 4 should have the delay "label" on the top right hand corner such as tose reported in figure 5.

6) The measured duty ratio of the grating ridge to the structure period (ϒ), should be better clarified through an equation or in a figure.

7) Origin of LIPSS. The authors states that they originate as a result of interference between laser beam and surface plasmon polaritons (SPPs). However, SPPs is commonly used to explain LIPSS on metals and on dielectrics under high intensity illumination, and recently in a work by Zhang et al.,Coherence in ultrafast laser-induced periodic surface structures, Physical Review B 92.17 (2015): 174109., a new mechanism involving coherent superposition between the scattered near-field at the surface and the incident electromagnetic field was suggested. The work was later confirmed by Rudenko et al., Spontaneous periodic ordering on the surface and in the bulk of dielectrics irradiated by ultrafast laser: a shared electromagnetic origin, Scientific Reports 7 (2017), 12306. The authors should reconsider or very well explain why structures formed on 4H-SiC may be originated by the only interference between the laser beam and surface plasmon polaritons (SPPs).

Author Response

Responses to Referees Comments

We would like to thank the editor for giving us a chance to resubmit the paper, and also appreciate the referees for their insightful suggestions. The following is a point-by-point response to the three reviewers’ comments.

Reviewer #3

  • This paper by He et al. reports the fabrication of bi-dimensional (2D) nanostructures using three collinear and temporally delayed femtosecond (fs) laser pulses. Additionally, the authors demonstrate that it is possible to control the structures' geometry by altering the mutual time delays among them. The work is interesting and will be appreciated by the scientific community, nevertheless, it contains some paragraphs that are not written in a clear way and needs to be reworded before publication (e.g. page 2 - lines 63-65: “Especially, during these double-fs-laser-beam irradiation experiments on semiconductor surfaces, the formation of 2D-LIPSSs has a rarely report, and the morphology distribution of 1D-ripple surface structures mostly presents the low regularity”. Overall, the work is remarkable particularly in terms of the presented physical model and the possible technological impact of the nanostructures. I suggest publication only after addressing the following points:

Response: According to the reviewer’s suggestion, in the revised manuscript, we have rewrote the paragraphs (at lines 68-70, in page 2), which can be described in detail as follows:

“It should be pointed out that, the formation of 2D-LIPSSs under irradiation of double-fs-laser-beam on semiconductor surfaces was rarely reported, and the morphology distribution of 1D-ripple surface structures mostly presented the low regularity.”

  • Important information is missing in the abstract. The authors should provide relevant measurements about the structures morphology (e.g. periodicity) and also laser and processing parameters should be specified, (e.g. wavelength, pulse duration, repetition rate). Since in the introduction it is emphasized how this technique could replace electron-beam lithography, it is important to provide the reader with information about the periodicity of the structures.

Response: The referee’s comment is constructive. To address this criticism, in the revised manuscript, we have added the laser and processing parameters in the abstract, which can be described in detail as follows:

“Controlling laser-induced periodic surface structures on semiconductor materials is of significant importance for micro/nano-photonics. We here demonstrate a new approach to form the unusual structures on 4H-SiC crystal surface under irradiation of three collinear temporally delayed femtosecond laser (800 nm wavelength, 50 fs duration, 1 kHz repetition) with orthogonal linear polarizations. Different types of surface structures, two-dimensional arrays of square islands (670 nm periodicity) and one-dimensional ripple structures (678 nm periodicity) are found to uniformly distribute over the laser-exposed areas, both of which are remarkably featured by the low spatial frequency. By altering the time delay among three laser beams, we can flexibly control the transition between the two surface structures. The experimental results are well explained by a physical model of the thermally correlated actions among three laser-material interaction processes. This investigation provides a simple, flexible and controllable processing approach for large-scale assembly of complex functional nanostructures on bulk semiconductor materials.”

  • From the introduction it seems to be implied that the transition from HSL- to LSF- LIPSSs is possible only when using a double pulse irradiation, however the transition from fine to coarse ripples has already been reported both for 6H-SiC and 4H-SiC by: 1) S.H. Kim, K.H. Byun, I.B. Sohn, S.H. Jeong, Progressive formation of fine and coarse ripples on SiC surface by repeated irradiation of femtosecond laser pulses, Applied Physics B 113 (2013) 395-402; 2) T. Tomita, K. Kinoshita, S. Matsuo, S. Hashimoto, Distinct Fine and Coarse Ripples on 4H-SiC Single Crystal Induced by Femtosecond Laser Irradiation, Japanese Journal of Applied Physics 45 (2006) L444-L446. For the importance of LIPSS of such systems and describe the significant differences in scales of the created nanostructures and what laser parameters might affect the self-organization process it is strongly suggested to consider the following very recent paper published and References reported therein: Mastellone et al., LIPSS Applied to Wide Bandgap Semiconductors and Dielectrics: Assessment and Future Perspectives, Materials 15 (2022) 1378. It should be stressed that the switch from LSF to HSF can be tuned by even a single pulse irradiation.

Response: According to the reviewer’s suggestion. In the revised manuscript, we have added some new representation and references [16], [17], [18] in the introduction part, which can be described in detail (at lines 43-47, in page 2) as follows:

“In general, the 1D ripple structures often presents two features: the high-spatial-frequency (HSF) and the low-spatial-frequency (LSF). And the switching from LSF to HSF cases can be tuned by a single laser pulse irradiation on semiconductors and dielectrics [16-18], but its controlling flexibility is much poor, which is mainly manifested as a self-organizing behavior.”

[16] Kim, S.H.; Byun, K.H,; Sohn, I.B.; Jeong, S.H. Progressive formation of fine and coarse ripples on SiC surface by repeated irradiation of femtosecond laser pulses. Appl. Phys. B 2013, 113, 395-402.

[17] Tomita, T.; Kinoshita, K.; Matsuo, S.; Hashimoto. Distinct fine and coarse ripples on 4H–SiC single crystal induced by femtosecond laser irradiation. JPN. J. Appl. Phys. 2014, 45, L444-L446.

[18] Mastellone, M.; Pace, M.L.; Curcio, M.; Caggiano, N.; Bonis, A.; Teghil, R.; Dolce, P.; Mollica, D.; Orlando, S.; Santagata, A.; Serpente, V.; Bellucci, A.; Girolami, M.; Polini, R.; Trucchi, D.M. LIPSS applied to wide bandgap semiconductors and dielectrics: assessment and future perspectives. Materials 2022, 15, 1378.

  • In the introduction are reported examples of regular 2D structures obtained on metal surfaces (tungsten, molybdenum and nickel). It is important to also show that it is indeed possible to obtain 2D structures on semiconductors or even transparent (non-absorbing) materials controlling the spatial period and the geometrical patterns of LIPSSs: 1) Wu, X., Jia, T., Zhao, F. et al. Formation mechanisms of uniform arrays of periodic nanoparticles and nanoripples on 6H-SiC crystal surface induced by femtosecond laser ablation, Applied Physics A 86, 491-495 (2007), 2) Mastellone, M. et al., Deep-Subwavelength 2D Periodic Surface Nanostructures on Diamond by Double-Pulse Femtosecond Laser Irradiation, Nano Letters 21 (2021) 4477-4483. By reporting these works it could be explained why a three-pulse sequence is preferable and how the formation mechanism differs.

Response: We agree with the reviewer’s suggestion. In the revised manuscript, we have added some new representation and the above mentioned research papers as new references [27], [28] in the introduction, which can be described in detail (at lines 57-60, in page 2) as follows:

“The recent investigations of the such a technique on different metal surfaces (Tungsten, Molybdenum, and Nickel) and semiconductors (Silicon carbide, Diamond) or even transparent materials demonstrated that several new types of 2D-LIPSSs, consisting of nano-bump, triangular, dot arrays, have been successfully created [25-28].”

[27] Wu, X.; Jia, T.; Zhao, F.; Huang, M.; Xu, N.; Kuroda, H.; Xu, Z. Formation mechanisms of uniform arrays of periodic nanoparticles and nanoripples on 6H-SiC crystal surface induced by femtosecond laser ablation. App. Phys. A 2007, 86, 491-495.

[28] Mastellone, M.; Bellucci A.; Girolami, M.; Serpent, V.; Polini, R.; Orlando, S.; Santagata, S.; Sani, E.; Fitzel, F.; Trucchi, D.M. Deep-subwavelength 2D periodic surface nanostructures on diamond by double-pulse femtosecond laser irradiation, Nano Letters 2021, 21, 4477-4483.

  • In the manuscript it is not that clear that the energy fluence used for the 1-pulse experiment (0.24 J/cm2) is different than the one used the 3-pulse irradiation (0.07 J/cm2). This is just reported in the caption of Fig.3 and it should be clearly explicated in the text.

Response: According to the reviewer’s suggestion. In the revised manuscript, we have added the energy fluence in the whole text.

  • Figures 3a and 4 should have the delay "label" on the top right hand corner such as tose reported in figure 5.

Response: To address this criticism, in the revised manuscript, we have added the delay “label” on the top right hand corner in figures 3a and 4.

  • The measured duty ratio of the grating ridge to the structure period (ϒ), should be better clarified through an equation or in a figure.

Response: To address the reviewer’s comments, in the revised manuscript, we have added a new sentence to clarify the above suggestion, which can be described, in detail (at lines 165-168, in page 5) as follows:

“The measured duty ratio of the grating ridge to the structure period approximates ϒ = 0.71 (ϒ = D / Λ, where D is the width of the grating ridge), which indicates the ablation groove width is about W = 203 nm, and the HSF ripple structures usually has the value of ϒ = 0.9.”

  • Origin of LIPSS. The authors states that they originate as a result of interference between laser beam and surface plasmon polaritons (SPPs). However, SPPs is commonly used to explain LIPSS on metals and on dielectrics under high intensity illumination, and recently in a work by Zhang et al.,Coherence in ultrafast laser-induced periodic surface structures, Physical Review B 92.17 (2015): 174109., a new mechanism involving coherent superposition between the scattered near-field at the surface and the incident electromagnetic field was suggested. The work was later confirmed by Rudenko et al., Spontaneous periodic ordering on the surface and in the bulk of dielectrics irradiated by ultrafast laser: a shared electromagnetic origin, Scientific Reports 7 (2017), 12306. The authors should reconsider or very well explain why structures formed on 4H-SiC may be originated by the only interference between the laser beam and surface plasmon polaritons (SPPs).

Response: The referee’s comments are constructive. It is well known that upon irradiation of the femtosecond laser pulse, a thin layer of plasma would like to transiently produce on the semiconductor surface because of the laser-excited free electrons via multiphoton absorption [S5,S6]. And if the electron density is above the critical value of Neth, the laser-exposed material surface tends to switch into the plasmonically active condition, which allows a direct excitation of SPP-like metallic state [S7]. Subsequently, the laser-SPP interference can produce the spatially periodic energy depositions to transiently modulate the material optical properties, leading to the formation of the permanent periodic structures on the sample surface. In other words, the photoexcited semiconductors can support the SPP propagation under irradiation of the high intensity femtosecond laser pulse. The surface of metallic state exists when the laser fluence is above the threshold of SPP excitation.

Moreover, the observed dependence of the ripple orientation on the laser polarization indicates the role of surface plasma. It can be understood that when the spatially fringe-like laser energy deposition patterns triggered by the SPP excitations with the high enough energy exceeding the material deformation threshold, the periodic grooves can be permanently imprinted on the material surface. It is hoped that the above physical understanding can promote the elucidation of the LIPSS phenomena on semiconductor materials. The further quantitative calculation of the above physical processes is under our investigations, and it will be shown in the future.

[S5] Obara, G.; Shimizu, H.; Enami, T.; Mazur, E.; Terakawa, M.; Obara, M. Growth of high spatial frequency periodic ripple structures on SiC crystal surfaces irradiated with successive femtosecond laser pulses. Opt. Express 2013, 21, 26323.

[S6] Derrien, T.J.-Y.; Krüger, J.; Itina, T.E.; Höhm, S.; Rosenfeld, A.; Bonse, J. Rippled area formed by surface plasmon polaritons upon femtosecond laser double-pulse irradiation of silicon. Opt. Express 2013, 21, 29643–29655.

[S7] Huang, M.; Zhao, F.; Cheng, Y.; Xu, N.; Xu, Z. Origin of laser-induced near sub- wavelength ripples: interference between surface plasmons and incident laser. ACS Nano 2009, 3: 4062–70.

A list of changes

  1. At lines 14-25, in page 1, the laser and processing parameters have been added in the abstract, and the abstract has been rewrote:

“Controlling laser-induced periodic surface structures on semiconductor materials is of significant importance for micro/nano-photonics. We here demonstrate a new approach to form the unusual structures on 4H-SiC crystal surface under irradiation of three collinear temporally delayed femtosecond laser (800 nm wavelength, 50 fs duration, 1 kHz repetition) with orthogonal linear polarizations. Different types of surface structures, two-dimensional arrays of square islands (670 nm periodicity) and one-dimensional ripple structures (678 nm periodicity) are found to uniformly distribute over the laser-exposed areas, both of which are remarkably featured by the low spatial frequency. By altering the time delay among three laser beams, we can flexibly control the transition between the two surface structures. The experimental results are well explained by a physical model of the thermally correlated actions among three laser-material interaction processes. This investigation provides a simple, flexible and controllable processing approach for large-scale assembly of complex functional nanostructures on bulk semiconductor materials.”

  1. At lines 43-47, in page 2, three new references [16,17,18] have been added , and a new sentence has been added: “In general, the 1D ripple structures often presents two features: the high-spatial-frequency (HSF) and the low-spatial-frequency (LSF). And the switching from LSF to HSF cases can be tuned by a single laser pulse irradiation on semiconductors and dielectrics [16-18], but its controlling flexibility is much poor, which is mainly manifested as a self-organizing behavior.”
  2. At lines 57-60, in page 2, two new references [27,28] have been added , and a new sentence has been added:

“The recent investigations of the such a technique on different metal surfaces (Tungsten, Molybdenum, and Nickel) and semiconductors (Silicon carbide, Diamond) or even transparent materials demonstrated that several new types of 2D-LIPSSs, consisting of nano-bump, triangular, dot arrays, have been successfully created [25-28].”

  1. At lines 68-70, in page 2, a new sentence has been added: “It should be pointed out that, the formation of 2D-LIPSSs under irradiation of double-fs-laser-beam on semiconductor surfaces was rarely reported, and the morphology distribution of 1D-ripple surface structures mostly presented the low regularity.”
  2. At line 128, in page 4, laser fluence “07 J/cm2/pulse” has been added.
  3. At line 158, in page 5, laser fluence “07 J/cm2/pulse” has been added.
  4. At line 178, in page 6, laser fluence “07 J/cm2/pulse” has been added.
  5. We have added the delay “label” on the top right hand corner in figures 3a and 4.
  6. At lines 165-168, in page 5, a new sentence has been added: “The measured duty ratio of the grating ridge to the structure period approximates ϒ = 0.71 (ϒ = D / Λ, where D is the width of the grating ridge), which indicates the ablation groove width is about W = 203 nm, and the HSF ripple structures usually has the value of ϒ = 0.9.”

Round 2

Reviewer 3 Report

Indeed, the effort for improving the manuscript quality is evident, it deserved to be published in the present form.